# Multiscale Virtual Screening Optimization for Shotgun Drug Repurposing Using the CANDO Platform

**DOI:** 10.3390/molecules26092581

**Published:** 2021-04-28

**Authors:** Matthew L. Hudson, Ram Samudrala

**Affiliations:** Department of Biomedical Informatics, Jacobs School of Medicine and Biomedical Sciences, University at Buffalo, Buffalo, NY 14203, USA; mlhudson@buffalo.edu

**Keywords:** drug repurposing, virtual screening, multiscale, multitargeting, polypharmacology, computational biology, drug repositioning, structural bioinformatics, molecular docking, proteomic signature

## Abstract

Drug repurposing, the practice of utilizing existing drugs for novel clinical indications, has tremendous potential for improving human health outcomes and increasing therapeutic development efficiency. The goal of multi-disease multitarget drug repurposing, also known as shotgun drug repurposing, is to develop platforms that assess the therapeutic potential of each existing drug for every clinical indication. Our Computational Analysis of Novel Drug Opportunities (CANDO) platform for shotgun multitarget repurposing implements several pipelines for the large-scale modeling and simulation of interactions between comprehensive libraries of drugs/compounds and protein structures. In these pipelines, each drug is described by an interaction signature that is compared to all other signatures that are subsequently sorted and ranked based on similarity. Pipelines within the platform are benchmarked based on their ability to recover known drugs for all indications in our library, and predictions are generated based on the hypothesis that (novel) drugs with similar signatures may be repurposed for the same indication(s). The drug-protein interactions used to create the drug-proteome signatures may be determined by any screening or docking method, but the primary approach used thus far has been BANDOCK, our in-house bioanalytical or similarity docking protocol. In this study, we calculated drug-proteome interaction signatures using the publicly available molecular docking method Autodock Vina and created hybrid decision tree pipelines that combined our original bio- and chem-informatic approach with the goal of assessing and benchmarking their drug repurposing capabilities and performance. The hybrid decision tree pipeline outperformed the two docking-based pipelines from which it was synthesized, yielding an average indication accuracy of 13.3% at the top10 cutoff (the most stringent), relative to 10.9% and 7.1% for its constituent pipelines, and a random control accuracy of 2.2%. We demonstrate that docking-based virtual screening pipelines have unique performance characteristics and that the CANDO shotgun repurposing paradigm is not dependent on a specific docking method. Our results also provide further evidence that multiple CANDO pipelines can be synthesized to enhance drug repurposing predictive capability relative to their constituent pipelines. Overall, this study indicates that pipelines consisting of varied docking-based signature generation methods can capture unique and useful signals for accurate comparison of drug-proteome interaction signatures, leading to improvements in the benchmarking and predictive performance of the CANDO shotgun drug repurposing platform.

## 1. Introduction

### 1.1. Drug Repurposing

Pharmacological innovation reduces human mortality rates and provides substantial improvements to the quality of life [1]. Therapeutic compounds that have been discovered, lab tested preclinically, and evaluated for risks and efficacy in clinical trials are approved by regulatory bodies such as the United States FDA for specific indications [2]. Potential failures impose high opportunity costs, and the realities of market forces and investment distort the types of ailments for which treatments are pursued [3,4,5]. The rate of novel drug discovery has been slowing as costs have been increasing, illustrating the need for more efficient paradigms [6].

Drug discovery traditionally relies on screening a compound or set of compounds against a biological target, typically a protein for a specific indication. Generally, these approaches incorporate high-throughput in vitro compound screens [7] and/or cell-based assays [8] of candidates drawn from wet laboratory studies or computational screens of virtual representations of compounds and biological targets [9,10,11]. If promising in vitro leads are found, they undergo in vivo testing, eventually leading to approval for clinical use if they continue to demonstrate relative efficacy and safety [2]. Traditional drug discovery methods tend to be focused on a single target and indication [11,12]. However, drugs and other human-ingested compounds interact promiscuously with many proteins in the body [13,14]. These off-target interactions are responsible for side effects and the fact that one drug may be useful for treating multiple indications [15,16,17,18,19]. Single-target approaches may miss promising leads and potentially beneficial off-target side effects, while drugs that have already been discovered and vetted for safety may be used in novel treatment contexts.

Drug repurposing is the practice of finding new uses for existing drugs, taking advantage of prior safety, efficacy, and pharmacological knowledge and data [15]. Drug repurposing has the potential to arbitrarily increase the utility of the FDA-approved drug library [17,20,21], particularly via innovations such as multitarget drug repurposing [22,23]. Drug repurposing has yielded new uses for multiple drugs [15,17] and has demonstrated potential for the treatment of viral [24,25], bacterial [26], and complex indications such as cancer [17].

### 1.2. Computational Drug Repurposing Using Molecular Docking

Computational models that improve drug discovery and repurposing leverage rapidly increasing computer processing power and vast collections of preclinical (in vitro, in vivo) and clinical data [22]. Although there are a variety of computational approaches, the most relevant ones to this study are structure-based. Structure-based approaches focus on modeling/simulating the effects the three-dimensional (3D) structure of a compound may have on one or more macromolecules, typically protein structures [27]. Structure representations are based on data obtained from X-ray diffraction, NMR spectroscopy, cryogenic electron microscopy, and biochemical and biophysical simulation studies. These models may incorporate other features such as predicted protein-compound binding sites, simulations of the surrounding chemical environments, and the functional characteristics of protein structures.

Molecular docking models the three-dimensional (3D) interaction between small molecule compounds and macromolecular protein structures [28,29,30,31,32]. Typically, these simulations algorithmically calculate the optimal position and orientation of a compound structure that interacts with (or binds to) a particular region of a protein structure and its corresponding interaction strength, using physics-based [33] or knowledge-based [34,35] force fields or scoring functions. The characteristics of a correctly modeled compound-protein structure provide researchers insight into the biological implications of the interaction: for example, a researcher may infer that a signaling pathway may be interrupted if a particular protein were to be inhibited by the compound based on the strength of its binding energy [36]. Molecular docking is also useful when researching large sets of compounds and proteins [37]. By comparing the relative differences in interactions between protein-compound pairs, the researcher can rank and organize pairs according to the strength of their interaction score and/or their similarity to identify patterns that are apparent only when examining large sets with many possible combinations, which is difficult and expensive to do in in vitro or in vivo experiments [22,38]. Molecular docking techniques have varying performance advantages and limitations [39,40]; however, provided that docking approaches are used wisely in concert with other experimental techniques, they have the potential to be useful for drug repurposing, particularly in a large-scale context [38].

### 1.3. Shotgun Multitarget Multi-Disease Drug Repurposing Using the CANDO Platform

The Computational Analysis of Novel Drug Opportunities (CANDO) platform was developed to mitigate endemic problems in drug discovery and enable multitarget approaches to drug repurposing [22,38,41,42,43,44,45]. The CANDO platform is designed to provide insights about the holistic behavior of compounds interacting within complex biological systems, including how a compound behaves relative to other compounds, and is an extensible standardized framework for building and combining drug repurposing, discovery, and design simulation pipelines. The similarity of drug-protein interaction behavior between a small molecule drug/compound and its macromolecular environment is hypothesized to indicate the similarity of drug therapeutic function [22]. In traditional structure-based and ligand-based drug discovery, therapeutic similarity inferences are often based on molecular target similarity and compound similarity [46]. CANDO extends the similarity assumption principle to include holistic multiscale interaction similarity, that characterizes compounds by the nature of their interaction with entire proteomes and (eventually) interactomes [22,38]. Extending the interaction similarity frontier enables CANDO to account for the promiscuous nature of compound interaction within biological systems and characterize previously unconsidered therapeutic functions of existing approved drugs. CANDO pipelines are evaluated by a benchmarking protocol that examines the relative ranking of every drug for every indication with two or more approved drugs. Analyzing the relative ranking of approved drugs for each indication enables the evaluation of the effectiveness of the platform for recovering known information, comparing relative pipeline performance for particular indications, calculating the accuracy and precision for ranking approved drugs, and determining which components of the platform need improvement.

In this study, we set out to extend the CANDO platform with an additional molecular docking pipeline using the popular software AutoDock Vina [30] to determine whether the prior CANDO performance was dependent on a specific molecular docking protocol, how different molecular docking protocols affect CANDO’s performance, and whether hybridizing molecular docking pipelines yields improved performance, as we have previously observed combining structure- and ligand-based CANDO pipelines [43].

## 2. Results

### 2.1. Benchmarking Performance of the Different Pipelines

The performance of two new primary pipelines and a hybrid one in the CANDO platform was investigated and compared to those previously created, including a random control (see Figure 1 and Methods). The first is the Vina pipeline that uses the eponymous molecular docking program to screen the CANDO v1.5 3733 drug/compound library against a 134-protein subset of the full proteome library (Vina-134). Multiple binding sites for each protein were predicted and targeted for docking, and the strongest interaction scores were used to construct the drug-proteome signatures.

The second pipeline used was the default CANDO v1.5 pipeline restricted to the same 134 protein subset (v1.5-134). We generated a hybrid decision tree pipeline drawn from a combination of the Vina-134 and the v1.5-134 pipelines. For comparison, we examined the performance of these pipelines with respect to a random control and the v1.5 pipeline implemented with the full CANDO proteome library consisting of 46,784 protein structures (v1.5-full).

Figure 2 illustrates the relative performance of these different pipelines. At the top10 threshold, the hybrid decision tree yielded 13.3% accuracy, v1.5-134 10.9%, and Vina-134 7.11%. The v1.5-134 and v1.5-full pipelines outperformed the Vina pipeline, but the latter was able to substantially contribute to the superior performance of the hybrid pipeline. Notably, the hybrid decision tree pipeline outperformed the v1.5-full pipeline with a top10 accuracy of 12.8% with two orders of magnitude difference in the number of proteins used in the implementation of the pipeline (134 vs. 46,784).

### 2.2. Divergence in Indication Accuracy at Various Thresholds

Figure 3 illustrates the similarity and divergence of indication accuracy performance at various thresholds: i.e., instances where the Vina-134 pipeline outperforms the v1.5-134 pipeline, instances where the v1.5-134 pipeline outperforms the Vina-134 pipeline, instances where each pipeline yields the same indication accuracy, and instances where each pipeline yields zero percent accuracy. At the top10 threshold, the Vina-134 pipeline had 191 indications (about 13% of all indications) that outperformed the v1.5 pipeline, which had 363 indications outperform Vina-134 (about 25% of all indications). There were 885 equivalently performing indications (with 828 of them at zero percent accuracy) at the top10 cutoff. Overall, the divergence in relative performance increased as the thresholds became less stringent (the CANDO pipeline outperformance share began to decline slightly after the top5% threshold). v1.5 had a higher number of indications in which it outperformed Vina-134. After the top50 cutoff, the proportion of equivalent indication accuracies that were both zero relative to the total equivalent indication accuracies began to decline rapidly.

### 2.3. Net Differences in Indication Accuracy

Figure 4 elucidates the net differences in top10 accuracies between two pipelines (and the proportion of approved drugs recovered per indication in the top10) for 700 indications. With some notable exceptions, the v1.5-134 pipeline outperformed the Vina-134 pipeline in frequency and magnitude. On a per indication basis, as the total number of approved drugs decreased, the Vina-134 pipeline had a higher number of the outperforming indications in terms of frequency and magnitude.

### 2.4. Relative Pipeline Indication Accuracy

The pipelines differed at average indication accuracy thresholds and on a per indication basis. In some cases, a pipeline that performed worse overall may do better for a specific indication. Figure 3, Figure 4 and Figure 5 illustrate the overall divergence, the magnitude of divergence, and the threshold frequency distributions. On a per-indication basis, there was divergence in the relative indication accuracy at various cutoffs, both in terms of net difference in accuracy, recovery at a particular threshold, and the frequency of a particular indication being recovered at a particular interval. The divergence between the per-indication performance of each pipeline elucidated by Figure 2, Figure 3 and Figure 4 suggests that each pipeline should be used in conjunction with one another for maximum indication inclusivity and accuracy.

### 2.5. Comparison of the Pipeline Distribution of per Indication Accuracies

Figure 5 illustrates the distribution of indication accuracies by counting the frequency of each indication that falls within a certain accuracy range. The dissimilarity of pipeline distributions at each cutoff was assessed by applying the Kolmogorov–Smirnoff test. The v1.5 pipeline outperformed Vina-134 overall (which yielded a higher frequency of indications exceeding 50% accuracy).

### 2.6. Indication Accuracy Distribution

Figure 6 examines the distribution of indications to illustrate their relative performance within each pipeline. Pipelines can also be compared with symmetrical accuracy distribution charts, where individual pipeline accuracy is denoted along the x and y axes. Each point can represent a particular indication (e.g., one of the 1439 indications in the CANDO platform), a defined indication class (e.g., all 39 indications with the string “neoplasm”), or some other way of denoting indications (e.g., indications that occupy a particular branch of the Medical Subjects Heading (MeSH) classification [47] or those that are ontologically similar [48]). When pipelines reach accuracy consensus (or near consensus) for a particular indication (or indication grouping), the point falls on or close to the 45 degree symmetry line. These figures suggest that different pipelines had varying success in benchmarking performance on a per-indication basis. More rigorous clustering analysis, indication classification, and indication definition will yield deeper insight into the relative strengths of each pipeline.

### 2.7. Distribution of Individual Drug-Indication Pair Rankings

Appendix A plots every drug-indication pair and its corresponding rank within each pipeline. These suggest that there is some substantial ranking consensus between each pipeline, as well as substantial divergence. The distribution was plotted at linear and logarithmic scales to illustrate the density of approved drug-indication pair ranking consensus and divergence. There is a high density of drug-indication pairs that have relatively high ranking in each pipeline. There is also a high density of drug-indication pairs that have a significantly higher ranking in the v1.5-134 pipeline than the Vina-134 pipeline. As with pipeline per-indication accuracy divergence, further investigation into drug-indication pair divergence may help improve the performance of individual and hybrid pipelines, particularly in cases where one pipeline ranked a drug-indication pair substantially higher than the other one (e.g., top100 in one and bottom 50% in the other).

## 3. Discussion

### 3.1. Multiple Large-Scale Virtual Screening Pipelines

In this investigation, we hypothesized that distinct docking methods would yield distinct drug-proteome interaction signatures due to differing simulation implementation, and correspondingly differing performance for shotgun drug repurposing: BANDOCK, the default bioanalytical or similarity docking in CANDO is a knowledge-based template/comparative modeling protocol [22,38,41,44,45], and AutoDock is a more traditional molecular docking approach with physics-based force fields [30]. Including other molecular docking pipelines beyond the default pipeline implemented in the platform enabled us to evaluate whether or not CANDO as a platform was specifically dependent on the drug-proteome signature generation methodology implemented in the default pipeline.

Our results demonstrated that the CANDO platform was not dependent on a single pipeline implementation, and that combining different virtual screening pipelines can yield better performance relative to using the individual ones. On a platform level, the drug-proteome signature ranking and indication recovery paradigm was viable using more than one means of signature generation. On a pipeline level, the pipelines (the two large-scale virtual screening pipelines and the combined decision tree pipeline) each demonstrated varying degrees of performance and instances of unique signal capture.

The Vina-134 pipeline implemented in this study was viable in that it performed substantially better than the random control and performed at a significant fraction of the performance of the original default pipeline that utilized a bigger protein set. However, small protein libraries have been shown to perform relatively well, and some subsets of protein libraries performed better than others [44]. As previously demonstrated [43], hybrid pipelines can draw from the strengths of each constituent pipeline. As is the case here, the absolute performance of the Vina-134 pipeline was not the best, yet it substantially contributed to the higher performing hybrid pipeline.

### 3.2. Limitations and Future Work

Although some pipelines yielded superior signal over others in specific circumstances, precisely identifying why this occurred warrants further investigation. On a per-indication basis, it is possible to identify the superior performance of one pipeline over another (Figure 3 and Figure 4), but the MeSH indication classes were not precisely defined or had varying levels of specificity to one another. This issue will be addressed in the future through the use of more precisely defined indication mapping, for instance by using a realism-based ontology [48,49]. We are also using mathematical, statistical, and machine learning techniques to rigorously evaluate and enhance CANDO’s pipeline performance, as well as to identify clusters of drug-indication pair rankings when comparing different pipelines and methods [43,50], to yield insight into the ability of each pipeline to accurately recover known per-indication association information and make useful predictions for downstream prospective preclinical and clinical validation [42,51].

## 4. Materials and Methods

### 4.1. CANDO Platform and Pipeline Implementation

Figure 1 provides an overview of the CANDO platform and the particular pipeline implementations relevant to this study. The platform uses drug/compound and protein structure libraries curated from public sources and implements protocols for drug- and compound-proteome interaction signature generation, signature similarity calculation and sorting, assessing whether known drugs are ranked highly for the correct indications for single or hybrid pipelines (benchmarking), and generating novel putative drug candidates for specific indications (prediction). CANDO’s drug ranking pipelines have utility in many repurposing research contexts. For example, these pipelines can be used for lead generation for subsequent in vitro/in vivo testing and eventual off-label clinical use by physicians. By assessing the top ranking subset of drugs, a researcher or clinician can efficiently infer promising experimental or clinical drug candidates based on drugs ranked relatively to FDA-approved drug treatments and prior experimental evidence. For many clinical indications, CANDO pipelines are able to identify and highly rank FDA-approved drug treatments along with drugs that are FDA approved for other indications. Researchers can also infer associations between clinical indication classes, diseases, and biological pathways through the examination of indication-indication association networks connected by highly ranked drugs they have in common or other features of their respective compound-proteome signature. As illustrative examples of the broad uses of CANDO, Appendix A and Appendix A describe the indication-indication associations for a selection of MeSH neoplasm indications based on shared drugs ranked in the top10 in the Vina pipeline.

#### 4.1.1. Drug/Compound, Protein Structure, and Indication Library Curation

The default CANDO pipelines were implemented using bio-/chem-informatic docking protocols, where interactions were predicted from curated drug and protein libraries. The specific implementations and evolution of the libraries were reported extensively in several publications [22,38,41,44,45]. Briefly, the initial versions of CANDO (v1 and v1.5) incorporated 46,784 proteins and 2030 indication associations for 1439 drugs (out of 3733 compounds in total). Much of the data were drawn from the Protein Data Bank [52], the Food and Drug Administration, PubChem [53], the Comparative Toxicogenomics Database [54], DrugBank [55], protein structure modeling [56], and other sources.

The pipelines used in this study relied on curated sublibraries of the structures of 3733 drugs/compounds and 134 proteins and 13,746 drug-indication mappings, obtained from the same sources as above. We used the sublibraries to rapidly evaluate the utility of multiple molecular docking pipelines.

#### 4.1.2. Drug- and Compound-Proteome Interaction Signature Generation

A CANDO virtual screening pipeline simulates the interactions between all of its proteins and drugs/compounds, usually 3D structures, and is not dependent on any particular approach to accomplish this. These simulations generate proteomic similarity signatures (the vector of drug-protein interaction scores). The default CANDO platform pipelines generate drug-proteome interaction signatures using bioinformatic and cheminformatic docking protocols also described elsewhere extensively [22,38,41,44,45]. These signatures were compared for similarity and ranked. CANDO pipeline Version 1.5 [45] is a refinement of the original default pipeline [22,38,41,44] that uses near identical libraries, but improved interaction scoring [45]. We extended the drug- and compound-proteome interaction signature protocols to include the calculated binding energies generated by the program AutoDock Vina [30], as well as created hybrid pipelines combining molecular docking with bioanalytical/similarity docking (further details below).

#### 4.1.3. Drug- and Compound-Proteome Signature Similarity Calculation and Sorting

Broadly, the CANDO platform works by sorting every drug/compound relative to every other one based on their similarity and then uses known drug-indication associations to assess performance (Figure 2). Various pipelines implemented in CANDO generate drug-proteome interaction signatures for similarity sorting [22,38,43,44,45]. Underlying this platform is the core assumption that the similarity of drug interaction behavior across a proteome may be used to infer similarity in therapeutic function. The similarity between each drug and every other drug/compound is calculated using the root mean squared deviation of the individual interaction scores across a pair of drug-proteome interaction signatures [38].

Combined drug-proteome interaction signatures form an interaction matrix, with drugs along one axis and proteins on the other. These signatures are compared with one another and then ranked on a per-drug basis, and the quality of the resulting ranking is evaluated using the leave-one-out benchmarking protocol described below.

### 4.2. Benchmarking CANDO Platform Pipelines

Our benchmarking protocol calculates the performance for every indication with at least two approved drugs (1439 out of 3733 total) at various cutoffs, considering only the the top10 (abbreviated as “top10”), top25, top37 or 1%, top50, top100, top5%, top10%, and top50% of similarly ranked drugs. For each indication, the accuracy was derived from calculating how many known drugs mapped to that indication were “recovered” and highly ranked at various cutoffs.

We utilized three metrics to benchmark pipeline performance: average indication accuracy, pairwise accuracy, and coverage [22,38,41,44,45], all assessed at the different cutoffs. The average (mean) indication accuracy (%) is the average of all individual indication accuracies. The individual indication accuracy metric was calculated using the formula c/d×100, where *c* is a count of the number of times at least one approved drug for the indication was recovered within a particular cutoff and *d* is the total number of drugs approved for that indication. The other two benchmarking metrics were pairwise accuracy (weighted average of indication accuracies using the total number of approved drugs per indication) and coverage (number of indications that have an accuracy greater than zero).

### 4.3. New and Hybrid Pipelines

The pipelines examined in this study were derived from similarity ranking and benchmarking drug-proteome interaction signatures generated by large-scale bioinformatic and molecular docking. The CANDO platform is not limited to using docking-based virtual screening pipelines and has the potential to incorporate many different approaches to pipeline implementation and data sets (for example, ligand centric approaches have proven quite effective [43]).

#### 4.3.1. Virtual Screening Pipeline Using Autodock Vina

We used a small sublibrary (134 proteins) of the full CANDO proteome library to create the new molecular docking virtual screening pipeline due to computational constraints and also because we previously have shown that appropriately selected sublibraries of a similar size from the full library yield similar or better benchmarking performance [44]. We used the popular software AutoDock Vina Version 1.1.2 [30] for molecular docking of each protein structure against 3733 drugs/compounds from the CANDO v1.5 libraries. As with BANDOCK, we used COFACTOR [57] to predict binding sites, for binding search space size optimization [58], and used the strongest interaction score (lowest calculated binding energy) for each simulation from multiple sites. The best interaction score values for a drug-protein pair were used to generate the drug-proteome signatures.

#### 4.3.2. Decision Tree Pipeline

Prior CANDO platform investigations have demonstrated that multiple pipelines can be combined into a hybrid decision tree to maximize indication accuracy by drawing from pipelines that produce the best performance on a per-indication basis [43]. We used a similar approach in this investigation, using the pipeline that had the highest performance at the top10 cutoff.

### 4.4. Controls

We also compared the benchmarking performance of the pipelines to values obtained using a hypergeometric distribution that estimates the numerical probability of making a correct prediction by chance. This is one of the random control reference benchmarks used in the CANDO platform, the implementation of which was covered in detail in prior publications [43,45]. Benchmarking performance was also compared to the default pipeline implementations in CANDO Version 1 and Version 1.5 using the complete libraries.

## 5. Conclusions

Our results indicated that the utilization of multiple diverse docking-based virtual screening approaches in drug repurposing contexts such as the CANDO platform improves benchmarking performance. The Vina-134 pipeline performance indicated that the CANDO platform hypothesis of drug behavior similarity is not limited to the original bionalytical or similarity docking protocol BANDOCK for interaction signature generation. The hybrid decision tree pipeline performance provided further evidence that multiple signature generation pipelines may be combined to yield improved performance. Ongoing and future platform enhancement will incorporate multiple signature generation protocols and pipeline synthesis using AI/machine learning approaches to optimize performance. These improvements in turn will lead to greater predictive power and higher confidence in novel drug candidates generated for specific indications, which will be verified via prospective preclinical and clinical studies.

## Figures and Tables

**Figure 1 molecules-26-02581-f001:**
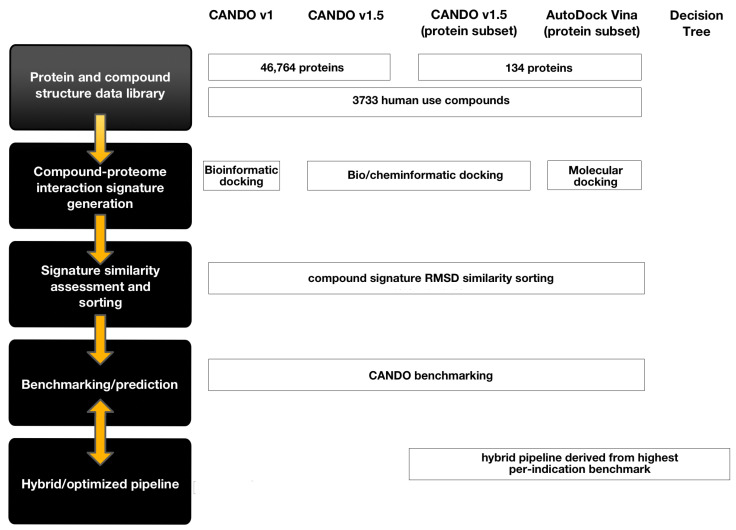
CANDO shotgun drug repurposing platform and pipeline overview. On the left side of the figure is a flow diagram, which indicates the general protocol for implementing a CANDO drug-proteome pipeline. To the right of the flow diagram, each pipeline relevant to this investigation is displayed along with implementation details for each phase of the CANDO protocol. Data curation: The drug-proteome pipelines utilize libraries of protein structure and drug structure representations. Interaction scoring protocol: These pipelines use bioinformatic, cheminformatic, and molecular docking methods to predict the scores between each protein and drug interaction. The set of protein interaction scores for each drug is considered its interaction signature. Each interaction signature can be compared with one another by assessing the root mean squared deviation of their interaction signatures. Drug comparison protocol: Every drug signature is compared with every other drug signature. After every comparison is made, each drug has a list containing the ordered set of every other drug, from most similar signature to least similar signature. Benchmarking protocol: The CANDO benchmarking procedure assesses, for every drug, how many other drugs with the same indication association are found within certain ranking cutoffs. An indication-specific accuracy score is produced by averaging the recovery rate of co-associated drugs for every drug associated with the particular indication for particular ranking cutoffs. The overall pipeline average indication accuracy is the mean of all the individual ones for a particular cutoff. Three pipelines were generated during this investigation: v1.5 (implemented with a subset of the CANDO proteome library), AutoDock Vina (using the same proteome sublibrary), and a hybrid decision tree pipeline derived from the former two pipelines. Each of the subset pipelines utilized a small sublibrary (134 proteins) of the original CANDO v1 and v1.5 pipelines. Although the pipelines used different signature generation approaches (BANDOCK bioanalytical or similarity docking and AutoDock Vina molecular docking), their signatures underwent the same similarity assessment and benchmarking protocol. However, there is room for variation via the use of alternate docking, similarity assessment, and benchmarking approaches.

**Figure 2 molecules-26-02581-f002:**
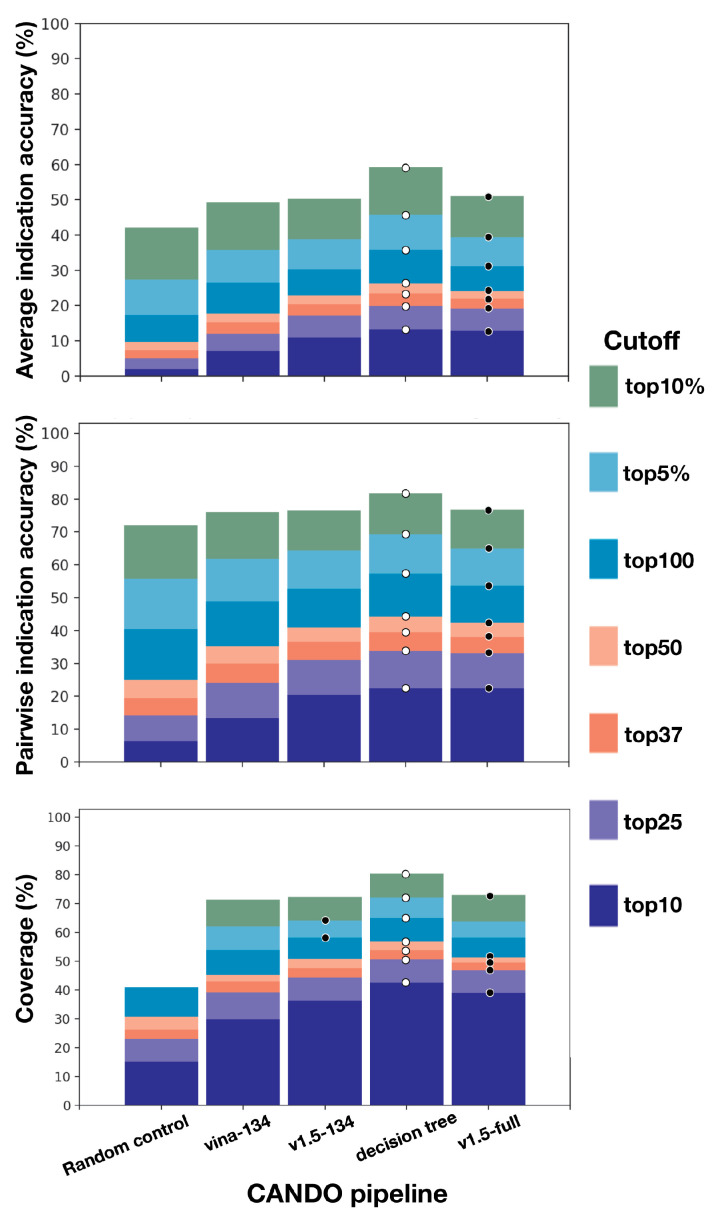
Benchmarking performance of CANDO pipelines used in this study. Three docking-derived pipelines implemented in CANDO: v1.5-full, using the interaction scores generated by the default CANDO v1.5 bioanalytical or similarity docking protocol (BANDOCK) for the full (46,784) proteome library, v1.5-134, using the same interaction scoring protocol for a 134 protein sublibrary, and Vina-134, based on interaction scores generated using AutoDock Vina for the 134 protein sublibrary are compared with a hybrid decision tree pipeline derived from combining the individual pipelines, as well as the random control reference pipeline calculated numerically from a hypergeometric distribution [43]. The pipelines were assessed by three CANDO platform benchmarking metrics: average indication accuracy (%), pairwise accuracy (%), and coverage (%). Performance cutoffs are denoted by colored bars from most to least stringent: top10 (dark purple), top25 (light purple), top37/top1% (dark pink), top50 (light)pink, top100 (dark blue), top5% (light blue), top10% (dark green), and top50% (light green) for 1439 indications with at least two approved drugs using a leave-one-out benchmarking protocol (see the Methods Section). White dots denote the highest overall accuracy at each threshold. The hybrid decision tree pipeline, which incorporates the highest indication accuracies from the Vina-134 and v1.5-134 pipelines, performed the best at all cutoffs (white dots). Black dots denote high performance in individual pipelines, which was obtained using the two v1.5 pipelines, one based on the 134 proteome sublibrary and the other on the full proteome library. v1.5-134 yielded the highest top50 percent average indication accuracy (85.3%), top100 coverage (52.742%), top5% coverage (64.382%), and top50% coverage (96.064%). Individually, the Vina-134 pipeline significantly outperformed the random control and yielded a significant fraction of the performance of the v1.5 pipelines. The hybrid decision tree pipeline performed the best, indicating that diversity in pipeline simulation implementation can be leveraged to increase drug repurposing performance.

**Figure 3 molecules-26-02581-f003:**
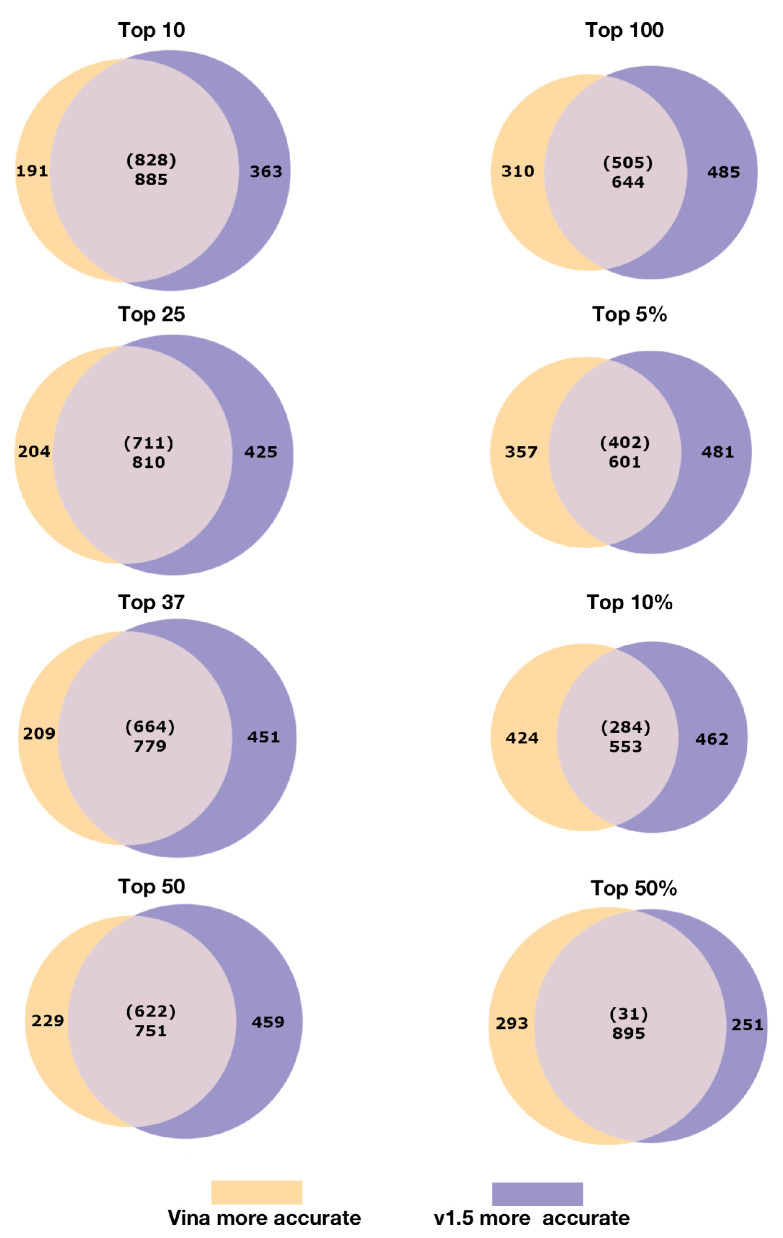
Comparing and contrasting indication accuracies for two CANDO platform pipelines at different cutoffs. Each Venn diagram represents the set of indication accuracies (1439 total) for the Vina-134 and v1.5-134 pipelines at different cutoffs (top10, top25, top37, top50, top100, top5%, top10%, and top50%). Indications that scored higher for the Vina pipeline are in yellow. Indications that scored higher for the v1.5 pipeline are in purple. Indications that scored the same for each pipeline are in gray. The number of indications where both pipelines yield 0% accuracy are provided in parentheses and the total number of equal indication accuracies provided below. At the top10 cutoff, the Vina-134 pipeline yields higher accuracies for 191 indications and the v1.5 pipeline for 363 indications, and each pipeline yields the same accuracy for 885 indications. Although the Vina-134 pipeline produces a substantial number of indication accuracies that are higher or equivalent to v1.5 indication accuracies, the v1.5 pipeline outperforms the Vina-134 pipeline at every cutoff except for top50%. Nonetheless, the orthogonality in the above diagrams indicates that individual pipelines can be synergistically combined into a hybrid pipeline that yields considerable performance improvements.

**Figure 4 molecules-26-02581-f004:**
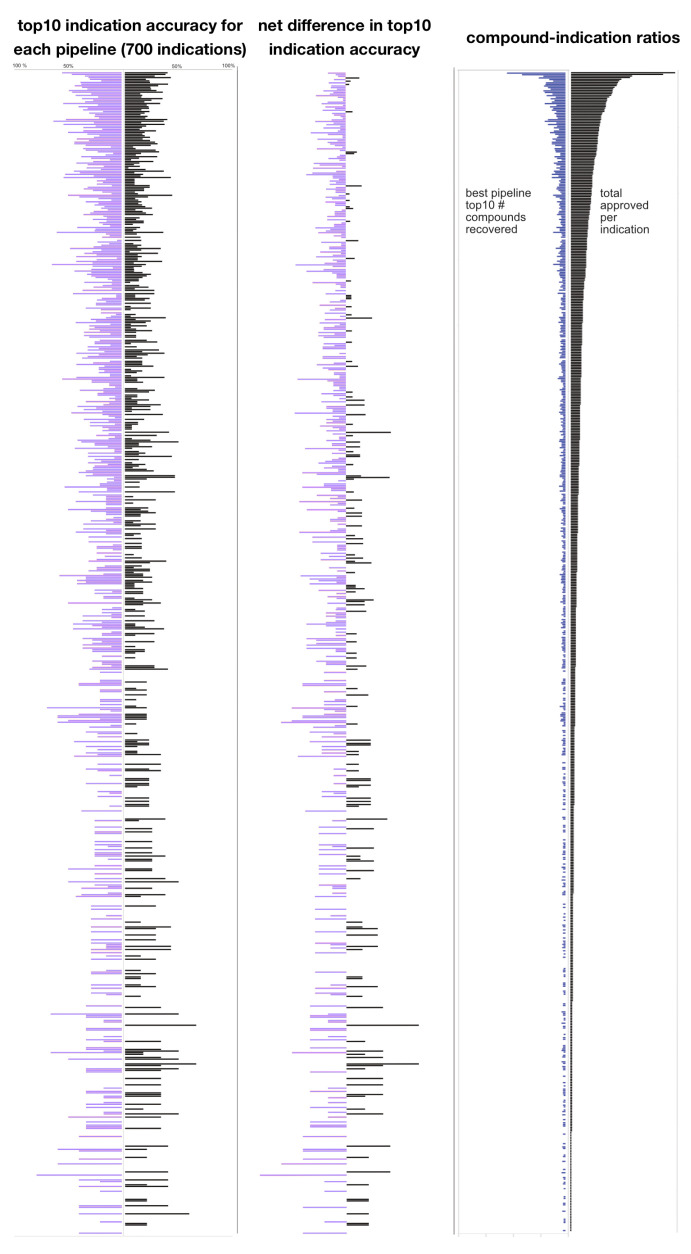
Comparison of 700 indication accuracies for two CANDO platform pipelines at the top10 cutoff. The top10 indication accuracies for 700 indications produced by the Vina-134 and v1.5-134 pipelines are shown in the left panel, with the the v1.5-134 pipeline per indication accuracies in purple on the left side and the Vina-134 pipeline accuracies in black on the right. The net difference in pipeline accuracy for the same indication is shown in the center panel, using the same percentage scale as the left. The number of drugs recovered by the best performing pipeline (in blue on the left side) and the total number of drugs approved per indication (in black on the right side) are shown in the right panel. The number of approved drugs for all three panels ranges from 158 drugs at the top to two drugs at the bottom. Generally, the v1.5-134 pipeline outperforms the Vina-134 pipeline, both by the number of indications and net difference in accuracy per indication.

**Figure 5 molecules-26-02581-f005:**
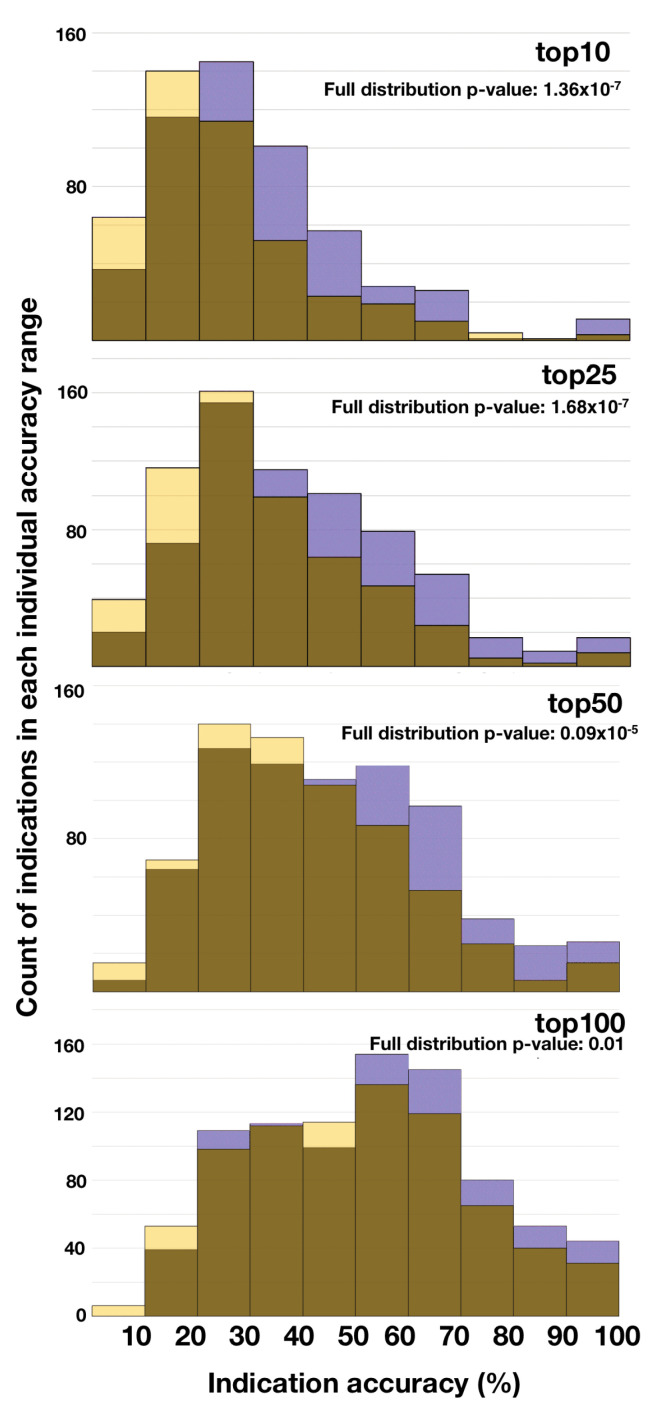
Frequency comparison of indication accuracies for two CANDO platform pipelines at different cutoffs. Shown are four histograms denoting the frequency with which indications fall within particular accuracy ranges (Vina-134 pipeline accuracies are in light yellow and v1.5 accuracies in black). The similarity of each distribution is assessed by the *p*-value using the Kolmogorov–Smirnoff test (*p*-values less than 0.05 are considered to be significant). The v1.5 pipeline outperforms Vina-134 overall, but the *p*-values indicate that the accuracy distributions are different for the two pipelines, indicating the utility of combining pipelines to produce synergistic performance.

**Figure 6 molecules-26-02581-f006:**
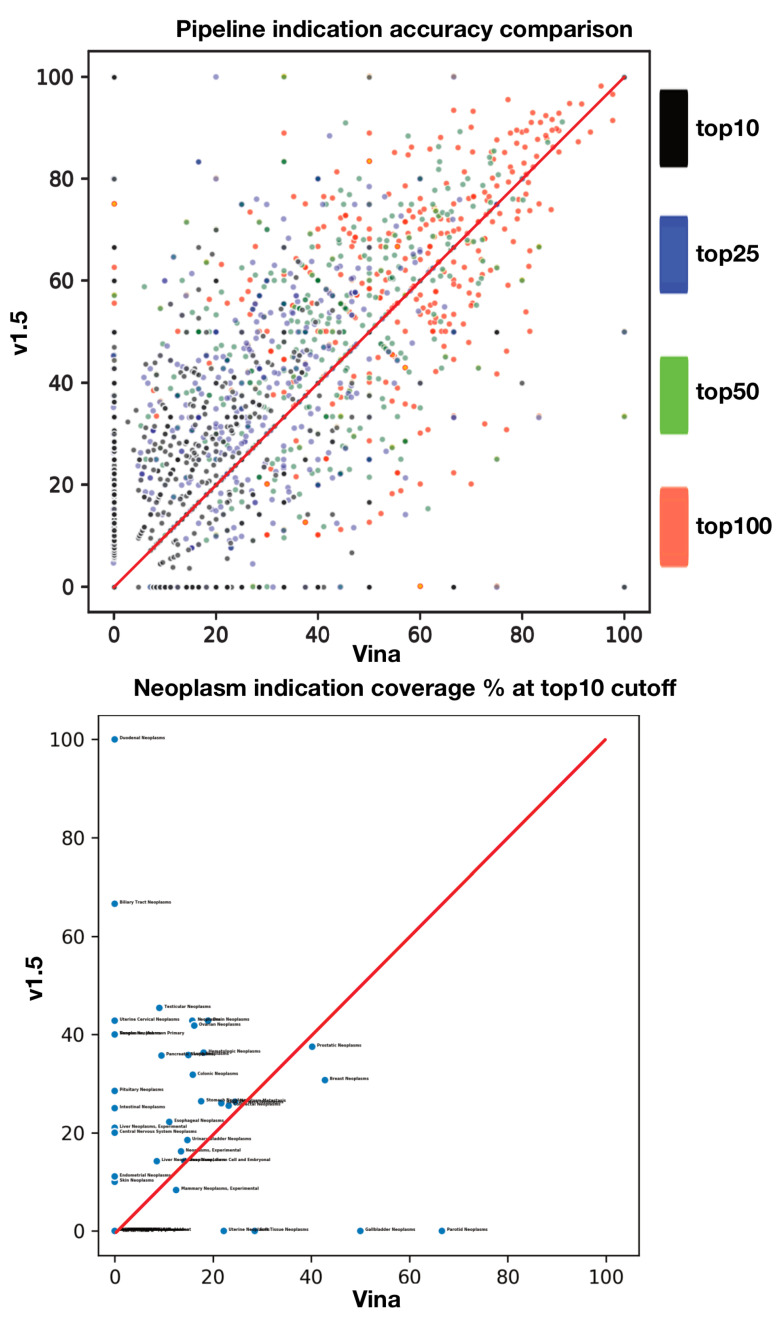
Comparison of the indication accuracies at various cutoffs and for a defined indication class for two CANDO platform pipelines. The top panel denotes a symmetrical accuracy chart. Each axis measures the indication accuracy for each pipeline, and indications are plotted according to their corresponding accuracies (different cutoffs are distributed in alternate colors). Points that land on the 45 degree red line are indications where the pipelines reached consensus; points that fall closer to a particular axis achieved a relatively higher score with the corresponding pipeline. The bottom panel isolates indications for the defined class “neoplasm” comprising 39 indications with the corresponding string. The asymmetrical distribution of the accuracy plot suggests pipeline accuracy differentiation, i.e., different pipelines have differing performance strengths and weaknesses, on a per-indication and indication class level.

## Data Availability

Publicly available datasets were analyzed in this study. This data can be found here: https://github.com/ram-compbio/CANDO and http://compbio.org/data/ both accessed on 27 April 2021.

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
