# Peer review of "Multiscale Virtual Screening Optimization for Shotgun Drug Repurposing Using the CANDO Platform"

_molecules, 2021, doi:10.3390/molecules26092581_

Round 1

Reviewer 1 Report

I checked the manuscript entitled: (Multiscale virtual screening optimization for shotgun drug repurposing using the CANDO platform) very carefully.

The authors claimed that docking based virtual screening pipelines have unique performance characteristics and that the CANDO shotgun repurposing paradigm is not dependent on a specific docking method. They also claimed that multiple CANDO pipelines can be synthesized to enhance drug repurposing predictive capability relative to their constituent pipelines.

In spite of the attractive soundness of such work and the important topic, this work remain pour theoretic and the information included will not give great help in drug discovery filed. In principle any theoretical results not supported clinically can be misleading. Also the manuscript contains some typos mistakes such as the one in line 88. 

Author Response

>>> "I checked the manuscript entitled: (Multiscale virtual screening optimization for shotgun drug repurposing using the CANDO platform) very carefully.

The authors claimed that docking based virtual screening pipelines have unique performance characteristics and that the CANDO shotgun repurposing paradigm is not dependent on a specific docking method. They also claimed that multiple CANDO pipelines can be synthesized to enhance drug repurposing predictive capability relative to their constituent pipelines.

In spite of the attractive soundness of such work and the important topic, this work remain pour theoretic and the information included will not give great help in drug discovery filed. In principle any theoretical results not supported clinically can be misleading. Also the manuscript contains some typos mistakes such as the one in line 88. "

We thank the reviewer for their time. We have edited the sentence on line 88 so it reads more clearly. As per the call for manuscripts and editor guidelines, this is a special issue on Computational Approaches, so that is why we did not include any prospective preclinical or clinical validation data. However the benchmarking performance evaluation presented in the manuscript is done using ap- proved drug data obtained from clinical trials, and the platform itself has been extensively validated in other publications. The goal here is to see if repurposing pipelines based on the popular Vina molecular docking method has a signal for drug repurposing that is greater than random and if there is orthogonality with existing CANDO pipelines.

Reviewer 2 Report

The article" Multiscale virtual screening optimization for shotgun
drug repurposing using the CANDO platform" present by the authors Matthew et al with the concept of  Computational Analysis of Novel Drug Opportunities (CANDO) platform is a high novelty and unique platform for drug discovery.

The paper has been written well with clear conclusions. Beyond the potentiality of the manuscript, I have a few minor concerns that the authors may address.

Minor: Consider the more recent studies in pharmacophore-based drug discovery in silico studies like PMID: 32499112, PMID: 29281938, PMID: 16712493, PMID: 26725317 are a more added value to this manuscript.

Author Response

>>> "The article" Multiscale virtual screening optimization for shotgun
drug repurposing using the CANDO platform" present by the authors Matthew et al with the concept of  Computational Analysis of Novel Drug Opportunities (CANDO) platform is a high novelty and unique platform for drug discovery.

The paper has been written well with clear conclusions. Beyond the potentiality of the manuscript, I have a few minor concerns that the authors may address.

Minor: Consider the more recent studies in pharmacophore-based drug discovery in silico studies like PMID: 32499112, PMID: 29281938, PMID: 16712493, PMID: 26725317 are a more added value to this manuscript."

We thank the reviewer for their time and literature suggestions. We have added PMID:29281938 as a reference.

Reviewer 3 Report

Samudrala et al. described a nice drug repurposing multiscale virtual screening using the CANDO platform. The interaction of a small molecule or drug with other proteins is often neglected even though it might have valuable insight. Having such examples in the literature (Sildenafil, Minoxidil, etc.) it is very important to understand the proteome-wide effect of the drug. Since the author is doing compound-proteome interaction signature generation (Fig 1 second box in the left), isn’t the docking of each compound with each protein is computationally expensive? As well as how the binding sites were established, or it was known? Are the source of CANDO v1 and v1.5 i.e. is it publicly available? It is not clear what all proteins were used and the detail about the curated library. It will be great to attach these files along with the manuscript. Why Autodock vina only selected among other software, was there any specific reason? The authors put great effort in showing various pipelines but didn't show how this can be useful in drug repurposing - can the authors show anyone a drug repurposing example or showing a wheel diagram for how a drug is connected to various diseases? 

In summary, the authors have put great effort into doing multiscale virtual screening optimization using the CANDO platform.  

Author Response

>>> "Samudrala et al. described a nice drug repurposing multiscale virtual screening using the CANDO platform. The interaction of a small molecule or drug with other proteins is often neglected even though it might have valuable insight. Having such examples in the literature (Sildenafil, Minoxidil, etc.) it is very important to understand the proteome-wide effect of the drug. Since the author is doing compound-proteome interaction signature generation (Fig 1 second box in the left), isn’t the docking of each compound with each protein is computationally expensive? As well as how the binding sites were established, or it was known? Are the source of CANDO v1 and v1.5 i.e. is it publicly available? It is not clear what all proteins were used and the detail about the curated library. It will be great to attach these files along with the manuscript. Why Autodock vina only selected among other software, was there any specific reason? The authors put great effort in showing various pipelines but didn't show how this can be useful in drug repurposing - can the authors show anyone a drug repurposing example or showing a wheel diagram for how a drug is connected to various diseases? 

In summary, the authors have put great effort into doing multiscale virtual screening optimization using the CANDO platform."

We thank the reviewer for their time and thoughtful response. Please see our detailed responses below.

1) RE: ”[...] isn’t the docking of each compound with each protein computationally expensive? [...]”

Indeed, proteome wide molecular docking is computationally expensive. However, compound-proteome signatures, once generated, can be quickly used and implemented into efficient drug repruposing pipelines (often amounting to a one-time fixed cost when generating entirely novel proteome sig- natures) which we have described previously in cited literature. Often, researchers do not need to regenerate compound-proteome signatures but instead use existing ones, such as those available via the CANDO platform (allowing them to avoid the computational cost of signature generation). We demonstrate that these signatures can be enhanced additively with additional signature data provided via an additional Vina-centric pipeline (which are in turn added to the signature pool available to researchers via the CANDO platform).

Should a researcher wish to replicate this approach on their own, signature generation and signature enhancement via alternate and/or hybrid pipelines would of course be dependent on their available processing power. If they do not have access to these resources, CANDO provides precomputed signatures and additional drug repurposing software that is efficiently useable on many common consumer grade laptops.

2) RE: ”[...] how were binding sites established? [...]”

We now describe our approach to binding site prediction and identification in the Methods section in further detail. (Briefly, we used COFACTOR and a search space optimization algorithm developed by another research group.)

3) RE: ”[...] CANDO publicly avaialable? [...] Which files?”

Yes all of the software and data that comprise the CANDO platform are publicly available. For a comprehensive overview, refer to http://protinfo.compbio.buffalo.edu/cando/. Users primarily use CANDO through the publicly available Python software library. This library also facilitates interacting with existing CANDO data (which can be downloaded locally or accessed via the internet). Many of the most popular features are available through the library and a (relatively) small accompanying database (users may prefer to make network requests to the public CANDO servers when accessing the larger set of CANDO data (e.g. they would like underlying structure files and other data). Users can also provide their own structure data and implement their own pipeline and use the Python library for benchmarking and other features (or even implement their own compound-proteome signature ranking pipeline and benchmarking algorithm using the underlying theoretical framework of CANDO separate from its current software implementation.)

This article describes the implementation and usage of the platform: https://pubs.acs.org/doi/10.1021/acs.jcim.0c00110.
The software is available at https://github.com/ram-compbio/CANDO

and

http://compbio.buffalo.edu/software/

where you can find more documentation about the API, data formats, and the latest release ver- sion. More information about the theory and implementation of CANDO is discussed extensively in the cited literature.

We find it more efficient to refer to the CANDO version number + protein library + compound library in use with a particular release version of the platform as there are thousands of files, IDs, and various corresponding metadata. These PDB IDs of the protein structure files and the names of the compounds are available on one of our public servers. In this study each pipeline drew from the protein and compound libraries used in version 1.5. We used the entire compound library and a subset of the protein library as input to each pipeline (a listing of the subset IDs are available at http://compbio.buffalo.edu/data/mc_cando_multiscale_optimization/).

4) RE: ”[...] why Vina? [...]”

Autodock Vina was selected because it is a widely used docking program which is familiar to many researchers who may be interested in this work. There are indeed many available molecular docking algorithms and software packages to choose from. We demonstrate that while some may perform better than others in head to head comparisons — hybrid approaches in these proteome-wide sig- nature generation pipeline contexts can enhance performance; so future work may include multiple docking programs beyond Autodock Vina in order to examine if additional docking program diversity is useful in increasing end-user drug repurposing prediction performance.

5) RE: ”[...] use for drug repurposing example? drug associations between diseases [...]”

We have added a figure and a table to the supplementary materials section depicting indication- indication associations. Also, we have added some discussion in the manuscript about how researchers use CANDO to facilitate drug repurposing research.

CANDO relative drug ranking pipelines have utility in many drug repurposing research contexts. For example, these pipelines can be used for lead generation for subsequent in vitro/in vivo testing and eventual off-label clinical use by physicians. By assessing the top ranking subset of drugs, a researcher or clinician can efficiently infer promising experimental or clinical drug candidates based on relative 

drug ranking to FDA approved drug treatments and prior experimental evidence. For many clinical indications, CANDO pipelines are able to identify and highly rank FDA approved drug treatments along with drugs that are FDA approved for other indications. Researchers can also infer associa- tions between clinical indication classes, diseases, and biological pathways through examination of indication-indication association networks connected by highly ranked drugs they have in common or other features of their respective compound-proteome signature. As illustrative examples of the broad uses of CANDO, Supplementary Figure 2 and Supplementary Table 1 describe the indication- indication associations for a selection of MeSH neoplasm indications based on shared drugs ranked in the Top 10 in the Vina pipeline. You can find these supplementary materials, as well as further information and data at http://compbio.buffalo.edu/data/mc_cando_multiscale_optimization/.

Round 2

Reviewer 1 Report

I can accept this article with concern due to the theoretical results of this work which is not supported by evidence.

This manuscript is a resubmission of an earlier submission. The following is a list of the peer review reports and author responses from that submission.